# Myoelectric, Myo-Oxygenation, and Myotonometry Changes during Robot-Assisted Bilateral Arm Exercises with Varying Resistances

**DOI:** 10.3390/s24041061

**Published:** 2024-02-06

**Authors:** Hsiao-Lung Chan, Ling-Fu Meng, Yung-An Kao, Ya-Ju Chang, Hao-Wei Chang, Szi-Wen Chen, Ching-Yi Wu

**Affiliations:** 1Department of Electrical Engineering, Chang Gung University, Taoyuan 33302, Taiwan; chanhl@mail.cgu.edu.tw (H.-L.C.); kya@mail.cgu.edu.tw (Y.-A.K.); t7036688@yahoo.com.tw (H.-W.C.); 2Neuroscience Research Center, Chang Gung Memorial Hospital, Linkou, Taoyuan 33305, Taiwan; chensw@cgu.edu.tw; 3Department of Occupational Therapy and Graduate Institute of Behavioral Sciences, Chang Gung University, Taoyuan 33302, Taiwan; lfmeng@mail.cgu.edu.tw; 4Division of Occupational Therapy, Department of Rehabilitation, Chiayi Chang Gung Memorial Hospital, Chiayi 61363, Taiwan; 5School of Physical Therapy and Graduate Institute of Rehabilitation Science, Chang Gung University, Taoyuan 33302, Taiwan; 6Department of Electronic Engineering, Chang Gung University, Taoyuan 33302, Taiwan; 7Department of Physical Medicine and Rehabilitation, Chang Gung Memorial Hospital, Linkou, Taoyuan 33305, Taiwan; 8Healthy Aging Research Center, Chang Gung University, Taoyuan 33302, Taiwan

**Keywords:** robot-assisted bilateral arm training, exercise intensity, muscle fatigue, mental fatigue, electromyogram, muscular oxygenation, myotonometry, heart rate

## Abstract

Robot-assisted bilateral arm training has demonstrated its effectiveness in improving motor function in individuals post-stroke, showing significant enhancements with increased repetitions. However, prolonged training sessions may lead to both mental and muscle fatigue. We conducted two types of robot-assisted bimanual wrist exercises on 16 healthy adults, separated by one week: long-duration, low-resistance workouts and short-duration, high-resistance exercises. Various measures, including surface electromyograms, near-infrared spectroscopy, heart rate, and the Borg Rating of Perceived Exertion scale, were employed to assess fatigue levels and the impacts of exercise intensity. High-resistance exercise resulted in a more pronounced decline in electromyogram median frequency and recruited a greater amount of hemoglobin, indicating increased muscle fatigue and a higher metabolic demand to cope with the intensified workload. Additionally, high-resistance exercise led to increased sympathetic activation and a greater sense of exertion. Conversely, engaging in low-resistance exercises proved beneficial for reducing post-exercise muscle stiffness and enhancing muscle elasticity. Choosing a low-resistance setting for robot-assisted wrist movements offers advantages by alleviating mental and physiological loads. The reduced training intensity can be further optimized by enabling extended exercise periods while maintaining an approximate dosage compared to high-resistance exercises.

## 1. Introduction

Upper-limb motor dysfunction is a prevalent symptom in individuals post-stroke. Bilateral arm training (BAT) involves repetitive, symmetrical arm movements and typically requires extensive guidance from a therapist. More recently, robot-assisted BAT has been introduced to assist with arm movement. This can be achieved through continuous passive motion (passive mode) or by using the unaffected arm to guide the movement of the affected arm, either with or without resistance (active mode) [1]. It has been shown that a recurrent and stable training procedure is effective in improving motor functions [2]. Furthermore, increasing the number of repetitions for robot-assisted BAT showed significant enhancements in motor function, muscle strength, performance of daily activities, and bimanual activity in individuals post-stroke [3].

Nonetheless, prolonged training sessions can result in both muscle and mental fatigue. Various measures, including surface electromyograms (EMGs), near-infrared spectroscopy (NIRS), heart rate monitoring, and the Borg Rating of Perceived Exertion (RPE) scale, have been employed to assess fatigue levels in conjunction with exercise. The Borg RPE serves as a frequently employed quantitative measure for gauging the subjective perception of effort during physical activity [4]. Demonstrating its reliability, the Borg RPE has shown a strong correlation with physiological markers such as heart rate and blood lactate concentration in the context of stepwise incremental exercise [5]. Additionally, the Borg RPE has been noted to increase concurrently with cardiorespiratory responses, as evidenced by elevated heart rate and oxygen uptake during graded arm cycling exercise in individuals with spinal cord injury [6].

EMG is widely recognized as the primary measure for evaluating muscle fatigue during exercise [7]. A shift towards lower frequencies in the EMG spectrum is a commonly used indicator to assess muscle fatigue during sustained isometric contractions [8,9,10,11,12]. The median frequency (MF) is a frequently employed parameter to characterize this spectral shift, with a reduction in MF considered an indicator of muscle fatigue. While isometric contractions provide a consistent state for assessing muscle fatigue, they require an additional procedure where participants are typically instructed to maintain a specific level of maximal voluntary contraction for assessment.

In one instance, a 30-s isometric contraction test was conducted between sessions involving repetitive robot-assisted anteflexion movements to assess muscle fatigue in individuals with multiple sclerosis [11]. Conversely, continuous EMG analysis throughout the exercise offers the advantages of ongoing fatigue assessment and establishes a direct link to movement dynamics, as demonstrated in previous studies [8,10,13]. Additionally, fatigue indicators are directly obtained from EMG while healthy subjects engage in robot-assisted upper-limb training, involving overcoming initial resistance and moving to a destination point [12].

As an alternative to myoelectric measurements, several studies have employed NIRS to monitor oxygen delivery and consumption during exhaustive muscle contractions. This approach has revealed a pattern of initial fast-phase oxygen desaturation, followed by a slower desaturation phase [14,15,16,17]. Additionally, decreases in tissue oxygenation levels have been documented in the forearm muscles during low-level voluntary contractions (less than 50% of maximal contraction) [18]. This phenomenon was also observed during 30% isometric contractions of the elbow flexors [19]. Furthermore, Felici et al. observed a more pronounced decline in MF of the EMG and a higher rate of oxygen desaturation in the biceps brachii muscle during sustained isometric contractions at higher intensity levels [15].

Exercise intensity can be heightened by either extending the training duration or by increasing the resistance level. Numerous studies have explored the impact of extending exercise duration [20,21,22] or elevating exercise resistance [22,23,24,25,26] on the physiological responses of participants. However, the physiological investigation into fatigue assessment has yet to be applied in the context of robot-assisted bimanual upper-limb exercises, and its impact on exercise intensity remains unexplored. Moreover, there is a growing interest in integrating various physiological measurements to assess exercise-induced responses and fatigue [15,20,26,27,28,29]. The inclusion of diverse indexes can elucidate the origins of exercise-induced fatigue, considering aspects of physiology, psychology, and cardiac adaptation.

In the present study, two types of robot-assisted bimanual wrist exercises were conducted: one involving long-duration, low-resistance workouts and the other featuring short-duration, high-resistance exercises. Each session was conducted one week apart. During the exercises, EMG, muscular tissue oxygenation, and heart rate monitoring were employed throughout the exercises to investigate muscular activation, oxygen consumption, and sympathetic nervous system activation, respectively. Concurrently, the Borg RPE scale was administered every minute to track participants’ self-perceived fatigue levels during the exercises. It is worth noting that myotonometry measurements were taken before exercise, immediately after, and during two subsequent post-tests to evaluate exercise-induced alterations in muscle tone, stiffness, and elasticity across different exercise intensities.

With a dedicated experimental design, the effects of exercises with varying intensity modalities on robot-assisted bilateral wrist movements can be elucidated. This examination considers both central fatigue (measured through Borg RPE and heart rate) and peripheral fatigue (assessed using EMG signals’ MF). The exploration extends to changes in muscle oxygen delivery and consumption, as well as muscular viscoelastic properties. Such insights could prove valuable for fine-tuning exercise intensity, optimizing rehabilitation efficacy, and minimizing fatigue occurrence.

The subsequent sections of this paper are organized as outlined below. The Materials and Methods section provides a comprehensive description of the experimental protocol, physiological measurements, and analysis methods employed in this study. The Results section presents the experimental findings and statistical comparisons. Ultimately, the Discussion and Conclusion sections provide a thorough exploration of the results in relation to prior works, address limitations, and outline potential future research directions.

## 2. Materials and Methods

### 2.1. Participants

Sixteen healthy right-handed male participants (aged 20.4 ± 2.1 years, with an average height of 173.5 ± 4.4 cm and weight of 64.8 ± 6.4 kg) were enrolled to engage in recurrent exercises using a robot-assisted bilateral arm trainer known as the Bi-Manu-Track, developed by Reha-Stim Co. in Berlin, Germany. Each participant was seated at a height-adjustable table, where they gripped 3 cm diameter handles and performed repeated wrist pronation and supination movements. To enhance engagement and motivation, a computer game involving the task of picking up and placing apples (Reha-Stim Co. in Berlin, Germany) was employed.

The active-resisted mode was implemented to control arm movement. The Bi-Manu-Track can be operated on either the right or left arm, driving the movement of the opposite arm. As all participants are right-handed, this study utilized the right arm to drive the movement of the left arm while requiring participants to overcome a specific level of resistance. Each participant completed wrist pronation and supination exercises with varying resistance levels on two separate occasions, with a one-week interval between sessions. For the low-resistance exercise, the active power, inward power, and outward power were set at 40, 50, and 50, respectively, whereas for the high-resistance exercise, these values were adjusted to 80, 50, and 50, respectively. It is important to note that the maximum available power for each configuration was capped at 100. The inward and outward power represented the initial resistances participants had to overcome to initiate pronation and supination, respectively. The incorporation of elevated active power in the high-resistance exercise was intended to generate increased resistance during participants’ pronation-supination movements. The sequence of resistance levels was counterbalanced among the participants. This study’s protocol was granted approval by the Research Ethics Committee of the Chang Gung Medical Foundation (IRB#101-3597B), and all participants provided their informed consent.

### 2.2. Experiment Protocol

Each experimental session consisted of four stages: warm-up, pretest, exercise, and two post-tests.
(1)Warm-up: A 2-min warm-up involving wrist movements was conducted. During the warm-up, the active power was set at 20, while inward and outward power were both set to 0.(2)Pretest: A pretest was conducted 2 min later, during which the participant performed ten wrist movements with the same resistance settings as used in the exercise mode. This pretest was followed by myotonometry measurements. The combined duration of the pretest and the subsequent resting period was 5 min.(3)Exercise: Subsequently, the participant engaged in wrist movements for a duration of 20 min in the low-resistance mode or for 10 min in the high-resistance mode. To regulate movement speed, a series of 1.2 s beep sounds were employed as guidance. Following the exercise, myotonometry measurements were taken once more.(4)Post-tests: Following a 5-min rest, each participant underwent a post-test employing the same procedure as the pretest, which included 10 repetitions of wrist movements followed by myotonometry. After an additional 5-min rest period, the second post-test was conducted.

### 2.3. Data Collection

As depicted in Figure 1a, two pairs of bipolar electrodes, separated by 2 cm, were longitudinally affixed. Each electrode, with a diameter of 0.88 cm and an area of 0.60821 cm^2^, was positioned midway between the distal motor endplate and the distal tendon on both the right extensor carpi radialis and right flexor carpi radialis, following the guidelines recommended by the European project “Surface EMG for non-invasive assessment of muscles (SENIAM)” [30]. Additionally, a ground electrode was placed on the olecranon. The EMG signals were amplified using a biopotential front-end amplifier, offering a gain of 12, an input resistance of 500 MΩ, and a common-mode rejection ratio of 115 dB. The signals were subsequently digitized at a sampling rate of 1 kHz with 24-bit resolution, utilizing an ADS1294 analog-to-digital converter (Texas Instruments, Dallas, TX, USA).

Simultaneously, tri-axial accelerations of the grasp handle were recorded. These accelerations had a full-scale range of ±3.6 g and a sensitivity of 300 mV/g, and were captured using the ADXL330 accelerometer (Analog Device, Cambridge, MA, USA). The Lead I electrocardiogram was also amplified under the same specifications as the EMG amplifier and sampled at a rate of 500 Hz with the ADS1294R analog-to-digital converter (Texas Instruments, Dallas, TX, USA). All of these measured data were wirelessly transmitted to a notebook computer for real-time display and storage.

Tissue oxygenation levels in the muscles were assessed using two portable dual-wavelength near-infrared spectroscopy devices operating at 760 and 850 nm. Two NIRS sensors were longitudinally positioned midway on both the left extensor carpi radialis and left flexor carpi radialis. These measurements were made at a sampling rate of 10 Hz using the PortaLite system (Artinis Medical Systems, Gelderland, The Netherlands). The parameters obtained from a 3 cm distance between the light-emitting diode and the photodiode detector were selected. Figure 2 displays the oxygen saturation rate (StO_2_) along with the relative concentrations of oxyhemoglobin (HbO_2_) and deoxyhemoglobin (HHb) in the left flexor carpi radialis of a participant.

### 2.4. Myotonometry

Myotonometry was carried out immediately following exercise and the completion of 10 repetitions of wrist movements in the pretest and two post-tests using the MyotonePRO device (Myotone AS, Tallinn, Estonia). The measurement involved applying a minor mechanical impulse to the upper portion of the muscle belly of the left extensor carpi radialis and the left flexor carpi radialis individually. The parameters derived from the resulting damping oscillatory response were used as indicators for the muscle’s tone, elasticity, and stiffness. These myotonometry measurements were repeated five times at the same location, and the resulting parameters were individually averaged.

The myotonometric parameters encompass the oscillatory frequency, which is the inverse of the period between the maximal oscillatory peak and its subsequent peak (reflecting muscle tone); the logarithmic decrement from the maximal oscillatory amplitude to the subsequent oscillatory amplitude (inversely associated with elasticity); and the maximal oscillatory amplitude, multiplied by the mass of the testing end of the myometer and divided by the deepest tissue displacement of the testing end (representing stiffness).

Muscle tone is defined as the passive muscle tension arising from a muscle’s intrinsic viscoelastic properties in the absence of contractile activity [31,32]. Elasticity signifies the muscle’s ability to regain its original shape after the application of a minor force. This characteristic is essential for effectively utilizing muscle energy and enhancing blood circulation during physical exertion [32]. On the other hand, muscle stiffness refers to the muscle’s ability to resist deformation caused by external force. Muscle stiffness has been shown to correlate with muscle tone [33]. Both stiffness and tone are commonly incorporated into the clinical assessment of spasticity [31].

### 2.5. Borg Rating of Perceived Exertion

While the participant was engaged in either low-resistance or high-resistance exercise, the Borg RPE scale was utilized to assess their perceived level of exertion. This involved asking the participant to express how challenging they found the exercise, using a range of RPE scales (6, 7, 9, 11, 13, 15, 17, 19, and 20). Each value on the RPE scale represented different degrees of exertion, ranging from “no exertion at all” to “maximal exertion”, with specific descriptors such as extremely light, very light, light, somewhat hard, hard, very hard, and extremely hard.

### 2.6. Electromyographic Analysis

All data processing methods in this study were developed using MATLAB^®^ R2015b software (The MathWorks, Natick, MA, USA). To define the times of movement cycles, one-axial acceleration perpendicular to the radial direction of the grasp handle was employed. In order to enhance the cycling-related component, the acceleration data underwent filtering through an equiripple finite impulse response (FIR) lowpass filter, characterized by a 1 dB ripple with a cutoff frequency below 1 Hz and an 80 dB attenuation for frequencies above 4 Hz.

The EMG signals underwent two filtering processes: First, an equiripple FIR highpass filter was applied (with a 1 dB ripple, *f* > 10 Hz, and an 80 dB attenuation for *f* < 8 Hz) to eliminate low-frequency motion artifacts. Then, an equiripple FIR lowpass filter was used (with a 1 dB ripple, *f* < 350 Hz, and an 80 dB attenuation for *f* > 375 Hz) to eliminate high-frequency noise from the signals. All these digital filters were implemented using the Filter Design and Analysis Tool provided by the MATLAB R2015b software. Within each cycle, the filtered EMG data were analyzed using a discrete Fourier transform with zero padding extended to 2 s. MF was calculated as the frequency that divides the power spectral density *Pxx*(*f*) into two equal power spectra:$$\int_{0}^{\mathrm{MF}}P_{xx}(f)df=\int_{\mathrm{MF}}^{f_{s}/2}P_{xx}(f)df$$
where *f_s_* is the sampling frequency.

Figure 3 depicts the series of MF values derived from the EMG signals of the right extensor carpi radialis during the exercises performed by a participant. A notable decline in MF was observed, especially during high-resistance exercises. To analyze this decline, two regression lines were applied to fit the MF reduction within the first 5 and 10 min from the start of the exercise, respectively. A steeper regression slope indicates a more pronounced MF decrease. The selection of a 10-min duration was employed to quantify the decline in MF throughout the entire high-resistance exercise process. Meanwhile, shortening the duration to the initial 5 min aims to specifically characterize the short-term impact of exercise on the decline in MF. As shown in Figure 3, high-resistance exercise resulted in steeper regression slopes compared to low-resistance exercise.

Furthermore, the Fatigue Progress Measure (FPM) was utilized to quantify the cumulative degree of MF decline over time. FPM is defined as the number of processed cycles with an MF lower than the initial MF, divided by the total number of processed cycles [13]. The initial MF is determined as the average of the MFs from the first 25 cycles, which corresponds to approximately 1 min of data. FPM values were computed every 20 s. As depicted in Figure 3, higher FPM values were observed during high-resistance exercise compared to low-resistance exercise.

### 2.7. Statistical Analysis

A repeated measures analysis of variance was conducted to compare myotonometric parameters and muscular oxygenation variables separately among the pretest, shortly after exercise, and the two post-tests, with a significance level of α = 0.05 (SPSS Statistic 22, SPSS Inc., Chicago, IL, USA). The dependent variables included oscillatory frequency, decrement, stiffness, and oxygen saturation rate, as well as the relative concentrations of oxyhemoglobin and deoxyhemoglobin. Post hoc comparisons were performed using one-way ANOVA with the LSD test to assess differences among the various tests (α = 0.05).

The mean heart rate, Borg RPE, and FPM derived from the EMG of right upper extremity muscles, as well as muscular oxygenation variables measured every minute, were analyzed using repeated measures analysis and C-statistic analysis during high-resistance exercise, as well as during the former and latter stages of low-resistance exercise. Measures with *p* < 0.05 and Z-scores greater than 1.645 were considered significant in terms of both difference and trend. Furthermore, parameters were averaged separately for the early (2nd–4th min), middle (5th–7th min), and late (8th–10th min) periods, and then subjected to a one-sided paired *t*-test for comparison between low-resistance and high-resistance exercises.

## 3. Results

### 3.1. Impacts of Varying-Resistance Exercises on Myotonometry and Muscular Oxygenation

Table 1 presents the statistical comparisons of myotonometric parameters before and after wrist pronation-supination exercise. Notably, only the low-resistance exercise led to significant post-exercise changes in myotonometric parameters compared to the pretest. Specifically, the oscillatory frequency of the left flexor decreased shortly after exercise (SAE) and at the first post-test, while the stiffness of both the left flexor and left extensor decreased at SAE and the first post-test. Additionally, the oscillatory amplitude decrement of the left flexor decreased at the first and second post-tests.

Table 2 presents the statistical comparisons of muscular oxygenation parameters before and after wrist exercises. Each parameter represents the average measurement from the third minute after wrist movements. Significant post-exercise changes were observed during SAE and the two post-tests when comparing these parameters to the pretest. These changes included higher StO_2_ in both the left flexor and left extensor during high-resistance exercise, higher StO_2_ in the left extensor during low-resistance exercise, and higher relative HbO_2_ in both the left flexor and left extensor for both low- and high-resistance exercises. Additionally, it is worth noting that the StO_2_ and HbO_2_ levels in the left flexor were at their highest at SAE in the context of high-resistance exercise.

### 3.2. EMG Median Frequency, Heart Rate, and Borg RPE throughout the Exercise Period

Figure 4 depicts the Borg RPE, mean heart rate, and FPM for the right flexor and right extensor measured every minute during wrist exercise. These parameters consistently demonstrated a significant increasing trend throughout the high-resistance exercise as well as in the former stage of low-resistance exercise. Furthermore, in the latter stage of low-resistance exercise, the FPM for the right extensor continued to increase while the mean heart rate decreased.

Table 3 provides the statistical comparisons of Borg RPE, mean heart rate, and FPM between low- and high-resistance exercises, considering three distinct time periods: early (2nd–4th min), middle (5th–7th min), and late (8th–10th min). In each of these periods, Borg RPE, mean heart rate, and the FPMs of the right extensor and right flexor were significantly higher during high-resistance exercise compared to low-resistance exercise.

Table 4 outlines the statistical comparisons of the slopes of regression lines that model the cycle-dependent changes in MFs between low- and high-resistance exercises. It is notable that high-resistance exercise resulted in a more pronounced MF decline within the first 5 min in the right extensor, indicated by a more negative slope_5_, compared to low-resistance exercise.

Figure 5 illustrates the changes in StO_2_ and the relative hemoglobin concentrations of the left flexor and left extensor muscles at one-minute intervals during wrist exercise. In both low- and high-resistance exercises, there was a rapid decline in StO_2_ and HbO_2_ concentrations, as well as a quick increase in HHb concentration during the initial 2 min. Subsequently, the StO_2_ and HbO_2_ concentrations increased. Notably, the HbO_2_ concentration of the left extensor continued to increase in the latter stage of low-resistance exercise. During high-resistance exercise, the initial rapid increase in HHb was followed by an elevation in HHb for the left extensor and a reduction in HHb for the left flexor.

Table 5 displays the statistical comparisons of StO_2_ and the relative hemoglobin concentrations between low- and high-resistance exercises, considering three distinct time periods: early (2nd–4th min), middle (5th–7th min), and late (8th–10th min). Notably, the relative HHb and HbO_2_ in the left flexor were significantly higher during two periods of high-resistance exercise when compared to low-resistance exercise.

## 4. Discussion

The current study employed Borg RPE, heart rate, EMG, NIRS, and myotonometry to assess the impact of varying exercise intensities. These intensities included long-duration, low-resistance workouts and short-duration, high-resistance exercises, specifically robot-assisted bilateral wrist movements. The key findings and implications are summarized as follows:During high-resistance exercises, both the Borg RPE and heart rate were higher compared to low-resistance exercises, indicating an elevated sense of exertion and increased sympathetic activation, respectively.High-resistance exercise resulted in a more pronounced decrease in EMGs’ MF, indicating greater muscle fatigue compared to low-resistance exercise.High-resistance exercise required heightened oxygen delivery to the working muscles, as indicated by elevated relative concentrations of oxyhemoglobin and deoxyhemoglobin compared to low-resistance exercise. This suggests an increased metabolic demand during such activities.Low-resistance exercise resulted in decreased muscle stiffness and enhanced elasticity both immediately after exercise and during post-tests, as compared to the pretest. This implies that engaging in light exercise could be advantageous for promoting muscle relaxation and flexibility.

Experimenting with different exercise intensities on separate days offers the advantage of commencing each measurement with a fresh start, eliminating lingering effects from prior workouts. This methodology, drawn from our previous ergometer cycling studies, involves conducting two types of fatigue tests with an equivalent dosage on separate days: engaging in short-duration, high-resistance exercise until exhaustion, and undertaking long-duration, low-resistance exercise until exhaustion [22,34]. The choice of resistance level is pivotal in determining the balance between central and peripheral fatigue. Our observations revealed a more pronounced decay in twitch force (muscle fatigue) following high-resistance cycling, contrasting with low-resistance cycling, where a greater reduction in voluntary activation level (central fatigue) occurred after low-resistance cycling as opposed to high-resistance cycling [22]. Moreover, exercise resistance plays a mediating role in influencing brain activity and connectivity. Particularly, there is a notable enhancement in cortical activation and cortico-cortical coupling during high-resistance exercise [34]. Based on these findings, the present study applied this separate-day, varying-resistance experimental protocol to investigate the physiological responses linked to fatigue during robot-assisted wrist exercises.

During exercise, both slow-twitch and fast-twitch muscle fibers are recruited. Fast-twitch fibers produce brief bursts of muscle strength but tend to fatigue quickly, whereas slow-twitch fibers can sustain long-endurance muscle contractions. Surface EMG records the action potentials of both types of muscle fibers. When muscle fatigue occurs, there is a noticeable shift towards lower frequencies in the spectral distribution of EMG. This shift is frequently quantified using the MF and is commonly utilized as an indicator of muscle fatigue [35,36], particularly in cases of peripheral neuromuscular transmission failure [37,38]. Our recent study unveiled a more pronounced reduction in muscle force during the high-resistance mode compared to the low-resistance mode, indicating that exercises with higher resistance induce greater peripheral muscle fatigue than those with lower resistance. The results of the current study are consistent with our previous research, illustrating that varying resistance levels during cycling evoke distinct types of fatigue [22]. Furthermore, the current investigation established that this pattern persisted throughout the entire exercise duration. Specifically, a steeper decline in MF was observed during high-resistance exercise, suggesting that this type of exercise recruited more fast-twitch muscle fibers, which are prone to quicker fatigue. To the best of our knowledge, this is the first study to show that the resistance setting can induce different types of fatigue in robot-assisted exercise, thereby emphasizing its clinical significance in the application of this exercise modality.

MF has been commonly used as a real-time fatigue indicator [39], and it has been shown to respond to different modes of exercise, including eccentric and/or concentric exercise [40]. However, one challenge in assessing muscle fatigue using MF is the variability of the derived values. To address this issue, one approach is to fit the MF values over time using a linear regression model. The slope of the regression line has been shown to be relevant to the degree of muscle fatigue [8,10]. For instance, in pull-up exhaustion movements, the MFs of muscles like the brachioradialis and the teres major decay more rapidly than during isometric contractions when supporting body weight until exhaustion [8].

Another challenge is the potential for MF to fluctuate over time, particularly in situations involving alternating muscle activations during dynamic tasks like ergometer cycling and robot-assisted movements, where muscles collaborate. The variability in MF was partly attributed to the rapid recovery of neuromuscular transmission, as previous studies have demonstrated that MF recovers faster than force after fatigue [38]. To address the issue of MF fluctuation, one can quantify the decline in MF using the FPM, defined as the accumulated percentage of MF below the initial MF at the onset of exercise. A higher FPM is associated with more significant fatigue progression in the muscle [13].

The intensity of exercise typically results in different degrees of MF decline in the working muscles. In a study, the regression slope of biceps’ MF during high-workload rope pulling was found to be greater than that observed in a low-workload condition [10]. Another study has also highlighted the relationship between the degree of MF decline and the scale of exercise [8,10]. In the present study, it was observed that high-resistance exercise resulted in a more pronounced MF decline compared to low-resistance exercise, as evidenced by a steeper regression slope. Additionally, both levels of exercise resistance led to a rise in the FPM over time. However, high-resistance exercise consistently demonstrated a higher FPM throughout the entire duration compared to low-resistance exercise.

NIRS offers a convenient and noninvasive method for assessing oxygen utilization and delivery in working muscles during exercise and in the post-exercise recovery phase. Research has shown that exhausting knee-extension exercise or sustained isometric contractions lead to a greater reduction in oxygen saturation rate and oxyhemoglobin concentration when a higher level of maximal voluntary contraction is maintained [14,15]. In the current study, it was observed that oxygen saturation rate and oxyhemoglobin concentration decreased rapidly and then gradually increased following the commencement of low- or high-resistance exercise. Notably, the relative concentrations of oxyhemoglobin and deoxyhemoglobin in the left flexor were higher during high-resistance exercise compared to low-resistance exercise. This suggests that high-resistance exercise necessitates the recruitment of a greater amount of hemoglobin to meet the increased workload demands.

Increased oxyhemoglobin concentration and reduced deoxyhemoglobin concentration are commonly observed characteristics associated with post-exercise recovery [41]. To assess muscle recovery, post-exercise tests involving maximal voluntary contractions at various time points have been employed [42,43]. In the present study, hemodynamic data were collected shortly after exercise and during two subsequent post-tests to examine post-exercise recovery. As expected, post-exercise oxyhemoglobin concentration and oxygen saturation rate were higher than the pretest values. Notably, the oxyhemoglobin concentration and oxygen saturation rate in the left flexor were at their highest shortly after high-resistance exercise and gradually decreased during the post-tests. This observation could be attributed to the phenomenon of excess post-exercise oxygen consumption, which is related to the metabolic requirements after exercise [44].

Simultaneous measurement of EMG and NIRS has been employed in various studies to assess muscular activation and fatigue during exercise [15,17,45]. The alterations in MF and NIRS parameters both reflect changes in peripheral fatigue, with MF indicating neuromuscular propagation changes and NIRS reflecting alterations within muscle tissue. The current study observed more pronounced changes in high-resistance exercise compared to low-resistance exercise, confirming that high-resistance exercise challenges peripheral neuromuscular structures and, in other words, induces more peripheral fatigue. Additionally, the present study specifically selected the extensor carpi radialis and flexor carpi radialis muscles for myoelectric, tissue oxygenation measurements, and myotonometry. This choice is attributed to their involvement in wrist pronation and supination, making them pertinent to our investigation. Moreover, these muscles are situated close to the surface of the forearm, enabling practical and accurate measurement.

Myotonometry has been employed to assess the viscoelastic properties of muscles, including muscle tone, stiffness, and elasticity, in individuals affected by strokes [32,33,46]. Increased muscle stiffness has been observed in individuals with stroke, particularly those with restricted hypertonia [47] and chronic stroke [48], when compared to healthy subjects. Moreover, individuals with Parkinson’s disease have been observed to exhibit higher stiffness in the biceps brachii when compared to healthy controls [49]. Myotonometry has been employed as a tool to assess therapeutic efficacy in Parkinson’s disease individuals. For example, studies have shown that deep brain stimulation effectively reduces the stiffness of the musculus extensor digitorum following passive wrist movements [50]. Furthermore, there is a negative correlation between the dose of dopaminergic treatment and the stiffness of both the biceps brachii and brachialis [51].

Physical intervention has proven effective in improving muscle tone, reducing stiffness, and increasing elasticity. For instance, interval training on a cycloergometer has demonstrated promise in decreasing both muscle tone and stiffness of the biceps brachii, concurrently mitigating Parkinsonian rigidity [52]. The use of sling suspension has been found to efficiently reduce muscle tone and stiffness in the upper lumbar muscles of healthy women who maintained a static-prone position, as compared to the control group without sling suspension [53]. In the present study, shortly after low-resistance exercise and the subsequent post-tests, the stiffness of the left extensor and the left flexor, along with the muscle tone of the left flexor, showed a significant reduction. Moreover, there was an observed increase in the elasticity of the left flexor during the post-tests. Conversely, no significant change in these parameters was noted after high-resistance exercise. This suggests that low-resistance wrist pronation and supination exercises could be beneficial in reducing muscle tone and stiffness, while simultaneously enhancing muscle elasticity. These findings align with the positive impact of training on muscle tone and stiffness observed in previous studies [52,53].

The Borg RPE and mean heart rate both exhibited an increase from the early to middle to late stages in both low- and high-resistance exercises. This indicates that both exercise regimens employed in this study consistently induced a sustained feeling of fatigue and sympathetic nervous system activation [54] throughout the duration of the exercise. Notably, high-resistance exercise consistently resulted in higher Borg RPE scores and mean heart rates compared to low-resistance exercise at every stage. One might consider Borg RPE as a central fatigue indicator and propose that its changes occur in parallel with the alterations in heart rate. However, previous studies have demonstrated that the change in Borg RPE is only parallel to heart rate during the initial phase of exercise, and it has been shown to be influenced by both central and peripheral factors during fatigue [22,55]. The findings of our study may be explained by the fact that individuals experienced a greater sense of exertion and heightened sympathetic nervous activation during high-resistance exercises compared to low-resistance exercises. Nevertheless, the increase in Borg RPE feeling could not exclude the influence of peripheral fatigue.

Several studies have incorporated Borg RPE scores in conjunction with physiological responses derived from ECG, blood lactate, and EMG measurements during exercise. Borg RPE serves as a subjective measure of effort, and these investigations have consistently revealed strong correlations with objective fatigue indicators, including heart rate [5,6] and blood lactate [5] during incremental exercise. In the present study, participants experienced a shift in perceived exertion from ‘somewhat hard’ to ‘very hard’ as high-resistance exercise progressed, in contrast to the shift from ‘light’ to ‘hard’ observed during equivalent times in low-resistance exercise. Notably, perceived exertion remained consistent after the prolonged duration of low-resistance exercise. The mean heart rate showed an increasing trend over time in both low- and high-resistance exercises, but it decreased during the extended period of low-resistance exercise. Borg RPE and mean heart rate were consistently higher at every stage of high-resistance exercise compared to low-resistance exercise. These findings underscore the efficacy of Borg RPE and mean heart rate in assessing fatigue progression induced by resistance exercise.

Furthermore, the FPM demonstrated an elevation in the percentage of reduced MF counts in both the vastus lateralis and gastrocnemius muscles in correlation with the Borg RPE and heart rate during cycling exercise-induced fatigue [13]. These muscle fatigue indices, along with the cardiac stress index derived from heart rate, exhibited associations with the Borg RPE [29]. The current study further revealed a continuous elevation in FPM during both low- and high-resistance exercises, with consistently higher FPM values at each stage of high-resistance exercise compared to low-resistance exercise. This underscores its effectiveness in evaluating the progression of fatigue induced by resistance exercise.

However, the combination of Borg RPE with parameters from NIRS or myotonometry has not been explored in previous research. The current study unveiled an increased hemoglobin demand alongside elevated perceived exertion during high-resistance exercise as opposed to low-resistance exercise. Conversely, myotonometry is typically measured in the resting state, making it more suitable for delineating the impact of exercise on muscle viscoelastic properties and post-exercise recovery rather than assessing fatigue levels. These comprehensive perspectives not only illuminate the alignment of perceived exertion with objective physiological measurements but also offer valuable insights into the development of fatigue indicators for physiological monitoring and the assessment of training efficacy.

EMGs have been employed to establish muscle fatigue indices [56,57] and parameters for assessing muscle co-contraction [58,59,60] in robot-assisted rehabilitation. In a particular study, muscle fatigue was identified by analyzing patterns of MF peaks observed during rhythmic arm movements, leading to the adjustment of the admittance controller damping term in a hand rehabilitation robot [56]. Furthermore, a recent investigation introduced an intelligent protocol for wrist–forearm rehabilitation, making use of an exoskeleton robot. This innovative system incorporates torque estimation based on a biomechanical model of the wrist and forearm joints to assess the level of rehabilitation. Additionally, a support vector machine classifier is employed for muscle fatigue detection, utilizing the MF and average spectral power derived from EMG. This approach ensures that torque estimation is not compromised by the impact of muscle fatigue [57]. In the present study, muscle fatigue was assessed by the FPM, which can produce an immediate output for fatigue monitoring during exercise. In future research, the integration of muscle fatigue detection and co-contraction assessment holds the potential to enhance the monitoring of muscle activation status in post-stroke hemiplegia. This integration can contribute to maintaining effective training regimens and allow for the adjustment of robot settings, such as resistance and range of motion. This approach underscores the importance of preventing muscle injuries and optimizing rehabilitation outcomes.

The current study utilized a computer game featuring tasks of picking up and placing apples corresponding to wrist movements to enhance engagement and motivation. However, the integration of an interactive virtual environment into robot-assisted rehabilitation has garnered increasing research attention. These approaches provide an immersive vision-based game interface [61], as well as competitive and cooperative gameplay [62], with the goal of enhancing individual participation. Moreover, real-time presentation of virtual targets, movement feedback, and behavioral responses such as trial completion time and score offer immediate feedback to participants. Additionally, the movement information acquired during training can be leveraged to monitor rehabilitation progress [61]. The physiological monitoring, including the EMG-derived fatigue indicator, NIRS-derived metabolic demand, and ECG-derived cardiac index in the present study, holds potential for further development in extended reality with movement kinematic analysis in future research endeavors.

## 5. Conclusions

In this study, robot-assisted bimanual wrist exercises were conducted with two distinct resistance levels. Utilizing evidence from EMG, NIRS, and Borg RPE, we identified that robot-assisted exercise challenges various types of structures and exerts differing influences on muscle activation and the subjective feeling of exhaustion levels. Our findings underscore the significance of factoring in exercise intensity when designing robot-assisted rehabilitation programs. High-resistance exercise is associated with increased sympathetic nervous activation, a heightened sense of exertion, greater muscle fatigue, and an elevated metabolic demand to manage the intensified workload. Conversely, choosing a low-resistance setting in robot-assisted wrist movements provides benefits by reducing post-exercise muscle stiffness, enhancing muscle elasticity, and minimizing both physical and mental exertion. When embarking on extended training programs aimed at enhancing muscle strength, function, and recovery across diverse populations, including individuals with conditions such as strokes, Parkinson’s disease, multiple sclerosis, spinal cord injury, etc., it is crucial to consider the potential manifestations of fatigue (central, peripheral, or a combination). This necessitates the exploration of personalized exercise protocols tailored to optimize the advantages of training while concurrently minimizing fatigue and discomfort based on individual requirements and fitness levels. Furthermore, the present experiment involved having the dominant hand drive the movement of the non-dominant hand. It is important to note that this protocol introduces a limitation, as the relationship differs from the unaffected hand driving the affected hand in post-stroke hemiplegia. Further research is needed to elucidate exercise-induced fatigue in post-stroke hemiplegia. This requires employing an integrated analysis approach that incorporates various measurements. Additionally, combining robot-assisted exercises with other interventions, such as neuromuscular electrical stimulation, could potentially enhance rehabilitation outcomes and contribute to a more comprehensive understanding of fatigue in this specific population.

## Figures and Tables

**Figure 1 sensors-24-01061-f001:**
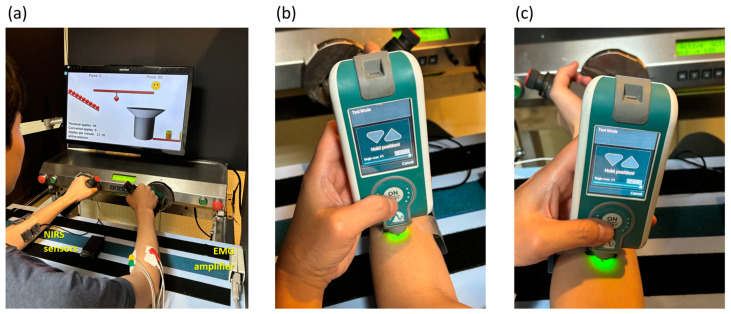
(**a**) During the experiment, a subject engaged in repetitive robot-assisted wrist pronation and supination movements using the Bi-Manu-Track, while being concurrently monitored with near-infrared spectroscopy (NIRS) and electromyogram (EMG) measurements on the left and right flexor carpi radialis and extensor carpi radialis. Resting myotonometry was performed on (**b**) the upper section of the muscle belly of the left extensor carpi radialis and (**c**) the corresponding area of the left flexor carpi radialis separately.

**Figure 2 sensors-24-01061-f002:**
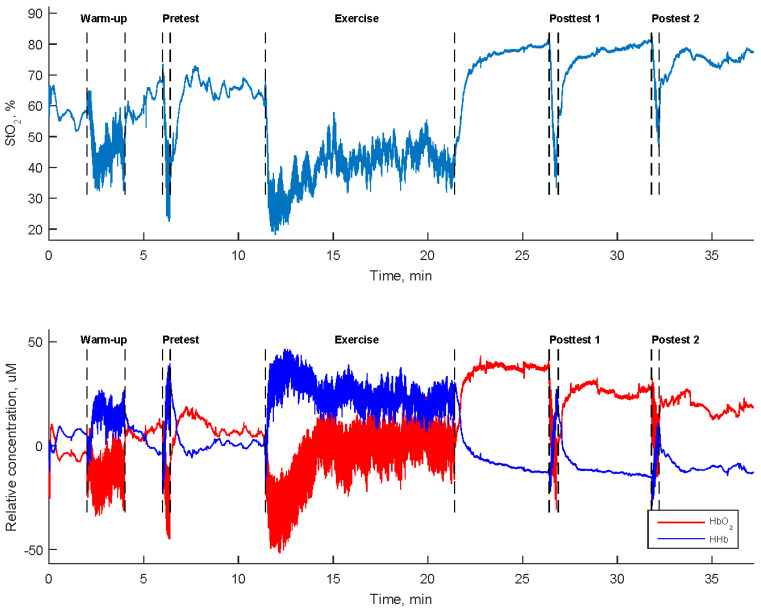
Tissue oxygen saturation rate (StO_2_) and the relative concentrations of oxyhemoglobin (HbO_2_) and deoxyhemoglobin (HHb) in the left flexor carpi radialis during a high-resistance bilateral robot-assisted wrist exercise carried out by a participant.

**Figure 3 sensors-24-01061-f003:**
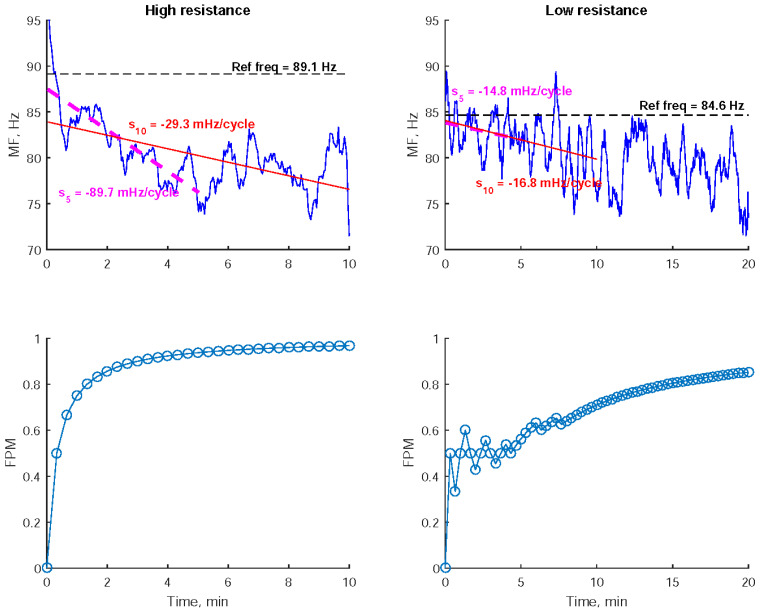
The 8-cycle moving–averaging median frequency (MF) of the electromyogram recorded from the right extensor carpi radialis during low- and high-resistance bilateral robot-assisted wrist exercises performed by a participant were examined. To quantify the decline in MF, the regression slopes within the first 5 and 10 min (S_5_ and S_10_), along with the Fatigue Progress Measure (FPM) were utilized. These metrics provided insights into the change in MF during the exercises.

**Figure 4 sensors-24-01061-f004:**
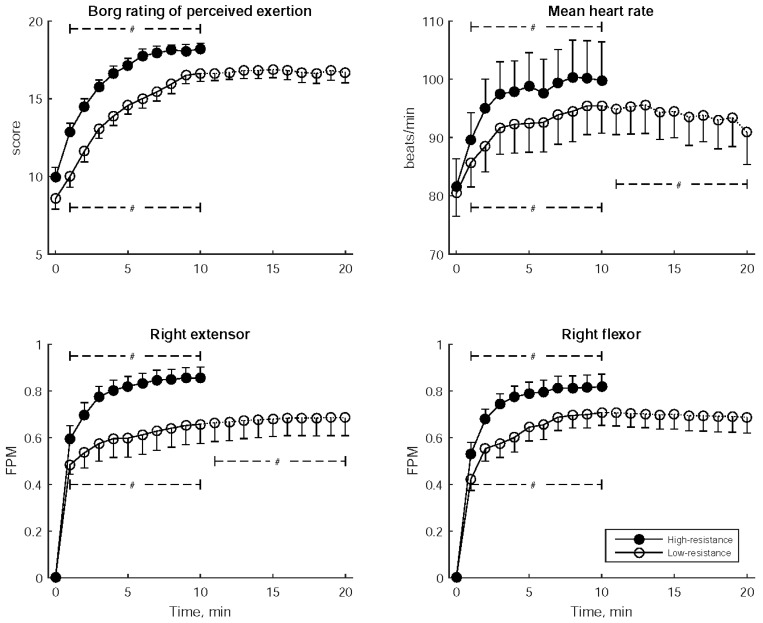
Borg Rating of Perceived Exertion, mean heart rate, and Fatigue Progress Measure (FPM) for both the right flexor and right extensor muscles, recorded every minute during bilateral robot-assisted wrist exercise. The data are presented as mean ± standard error of mean. A significant trend over the stages is indicated by # (*p* < 0.05).

**Figure 5 sensors-24-01061-f005:**
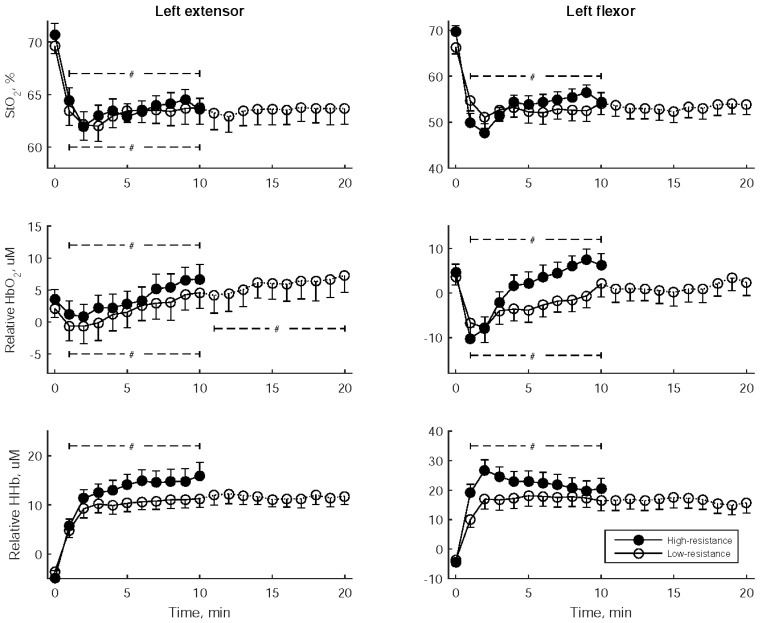
Tissue oxygen saturation (StO_2_) and the relative concentrations of oxyhemoglobin and deoxyhemoglobin (HbO_2_ and HHb) at one-minute intervals during bilateral robot-assisted wrist exercise. The data are presented as mean ± standard error of the mean. A significant trend over the stages is indicated by #.

**Table 1 sensors-24-01061-t001:** Myotonometric parameters before and after bilateral robot-assisted wrist exercise.

	Pretest	SAE	Post-Test_1_	Post-Test_2_	*p* Value
High resistance					
Left extensor					
Frequency, Hz	24.4 ± 2.6	23.7 ± 2.9	23.8 ± 1.7	23.6 ± 2.4	0.681
Decrement	1.41 ± 0.35	1.32 ± 0.29	1.40 ± 0.27	1.37 ± 0.33	0.619
Stiffness N/m	487.6 ± 45.5	475.5 ± 70.3	472.9 ± 34.2	477.6 ± 47.2	0.727
Left flexor					
Frequency, Hz	23.7 ± 3.1	23.1 ± 3.5	23.1 ± 2.6	23.5 ± 2.4	0.741
Decrement	1.96 ± 0.45	1.88 ± 0.33	1.95 ± 0.47	1.82 ± 0.41	0.188
Stiffness N/m	444.0 ± 33.1	446.0 ± 39.5	447.2 ± 39.2	451.7 ± 38.7	0.754
Low resistance					
Left extensor					
Frequency, Hz	23.8 ± 2.2	22.9 ± 3.3	22.5 ± 2.1	22.7 ± 2.7	0.214
Decrement	1.30 ± 0.21	1.26 ± 0.22	1.32 ± 0.31	1.30 ± 0.23	0.747
Stiffness N/m	483.1 ± 52.5	428.9 ± 68.3 ^†^	440.7 ± 64.4 ^†^	433.1 ± 58.2 ^†^	0.000
Left flexor					
Frequency, Hz	23.3 ± 3.2	21.9 ± 2.3 ^†^	21.4 ± 2.2 ^†^	22.3 ± 2.5	0.020
Decrement	1.85 ± 0.42	1.70 ± 0.44	1.62 ± 0.24 ^†^	1.64 ± 0.30 ^†^	0.020
Stiffness N/m	450.7 ± 53.1	422.5 ± 40.6 ^†^	420.6 ± 49.3 ^†^	434.5 ± 57.5	0.015

The data are presented as mean ± standard deviation, with SAE referring to shortly after exercise. Significance in the repeated measures analysis is denoted by the *p* value. The symbol † signifies a significant difference in post hoc comparisons when compared to the pretest (*p* < 0.05).

**Table 2 sensors-24-01061-t002:** Muscular oxygenation parameters before and after bilateral robot-assisted wrist exercise.

	Pretest	SAE	Post-Test_1_	Post-Test_2_	*p* Value
High resistance					
Left extensor					
StO_2_, %	68.4 ± 3.0	71.1 ± 4.4 ^†^	70.7 ± 3.7 ^†^	70.2 ± 4.1 ^†^	0.001
HbO_2_, μM	8.58 ± 7.61	22.17 ± 12.26 ^†^	21.20 ± 11.68 ^†^	18.68 ± 9.71 ^†^	0.000
HHb, μM	2.66 ± 2.66	3.44 ± 6.52	4.18 ± 8.15	3.04 ± 6.75	0.597
Left flexor					
StO_2_, %	67.5 ± 6.0	73.6 ± 6.5 ^†^	71.0 ± 7.3 ^†^	70.7 ± 5.8 ^†^*	0.000
HbO_2_, Μm	11.83 ± 8.81	29.78 ± 11.31 ^†^	22.71 ± 8.49 ^†^*	21.16 ± 7.57 ^†^*	0.000
HHb, μM	0.62 ± 5.00	−2.35 ± 5.14	−2.32 ± 6.18	−1.93 ± 5.66	0.073
Low resistance					
Left extensor					
StO_2_, %	68.1 ± 2.4	70.0 ± 3.9 ^†^	70.8 ± 3.8 ^†^	71.0 ± 4.3 ^†^	0.000
HbO_2_, μM	4.19 ± 6.71	18.48 ± 6.88 ^†^	18.11 ± 6.97 ^†^	17.67 ± 7.62 ^†^	0.000
HHb, μM	1.45 ± 3.80	2.84 ± 5.73	1.86 ± 4.81	0.60 ± 4.83	0.318
Left flexor					
StO_2_, %	63.9 ± 6.4	65.6 ± 6.2	65.5 ± 7.3	66.9 ± 6.2	0.193
HbO_2_, μM	5.87 ± 6.65	21.95 ± 9.98 ^†^	19.93 ± 8.29 ^†^	18.66 ± 6.76 ^†^	0.000
HHb, μM	1.18 ± 4.53	1.70 ± 5.81	−0.51 ± 5.48	−2.56 ± 5.78 ^†^	0.026

The data are presented as mean ± standard deviation. SAE refers to measurements taken shortly after exercise. StO_2_ stands for the oxygen saturation rate, and HbO_2_/HHb represents the relative concentration of oxyhemoglobin/deoxyhemoglobin. Significance in the repeated measures analysis is indicated by the *p* value. The symbols † and * denote significant differences in post hoc comparisons against the pretest and SAE, respectively (*p* < 0.05).

**Table 3 sensors-24-01061-t003:** Borg Rating of Perceived Exertion (RPE), mean heart rates, and Fatigue Progress Measures (FPM) during bilateral robot-assisted wrist exercises.

	Early (2nd–4th min)	Middle (5th–7th min)	Late (8th–10th min)
Borg RPE			
High resistance	15.6 ± 1.9 ^†^	17.6 ± 1.8 ^†^	18.1 ± 1.4 ^†^
Low resistance	12.9 ± 2.6	15.0 ± 2.3	16.4 ± 2.2
Mean heart rate, BPM			
High resistance	96.8 ± 18.3 ^†^	98.6 ± 20.0 ^†^	100.1 ± 22.5 ^†^
Low resistance	90.8 ± 16.0	93.0 ± 17.3	95.1 ± 17.1
FPM			
Right extensor			
High resistance	0.75 ± 0.20 ^†^	0.83 ± 0.18 ^†^	0.86 ± 0.19 ^†^
Low resistance	0.55 ± 0.30	0.59 ± 0.34	0.63 ± 0.34
Right flexor			
High resistance	0.74 ± 0.19 ^†^	0.80 ± 0.21 ^†^	0.81 ± 0.23 ^†^
Low resistance	0.60 ± 0.22	0.69 ± 0.24	0.72 ± 0.22

The data are presented as mean ± standard deviation, with BPM representing beats per minute. The symbol † is used to indicate a statistically significant difference between low- and high-resistance exercises (*p* < 0.05).

**Table 4 sensors-24-01061-t004:** Regression slopes of median frequency during bilateral robot-assisted wrist exercises.

	Slope_5_	Slope_10_
Right extensor		
High resistance	−96.5 ± 82.9 ^†^	−23.5 ± 32.1
Low resistance	−20.3 ± 72.2	−15.8 ± 22.8
Right flexor		
High resistance	−46.6 ± 45.9	−13.1 ± 19.9
Low resistance	−28.2 ± 40.4	−13.1 ± 17.6

The data are presented as mean ± standard deviation, and Slope_5_/Slope_10_ represents the slope of the regression line that fits the median frequencies within 5 or 10 min. The symbol † signifies a statistically significant difference between low- and high-resistance exercises (*p* < 0.05).

**Table 5 sensors-24-01061-t005:** Muscular oxygenation parameters during bilateral robot-assisted wrist exercises.

	Early (2nd–4th min)	Middle (5th–7th min)	Late (8th–10th min)
StO_2_, %			
Left extensor			
High resistance	62.8 ± 4.3	63.4 ± 3.9	64.1 ± 3.9
Low resistance	62.4 ± 5.1	63.5 ± 4.4	63.6 ± 5.7
Left flexor			
High resistance	51.1 ± 6.8	54.3 ± 6.8	55.4 ± 6.9
Low resistance	52.3 ± 8.7	52.4 ± 9.1	53.1 ± 8.9
HbO_2_, μM			
Left extensor			
High resistance	1.8 ± 7.5	3.8 ± 8.2	6.3 ± 8.1
Low resistance	0.1 ± 10.3	2.4 ± 9.3	4.0 ± 9.8
Left flexor			
High resistance	−2.8 ± 9.1	3.4 ± 10.0 ^†^	6.6 ± 9.3 ^†^
Low resistance	−5.2 ± 11.2	−2.8 ± 10.0	−0.1 ± 10.4
HHb, μM			
Left extensor			
High resistance	12.2 ± 7.1	14.5 ± 8.7	15.2 ± 10.0
Low resistance	9.7 ± 6.8	10.6 ± 6.4	11.1 ± 6.7
Left flexor			
High resistance	24.8 ± 12.9 ^†^	22.4 ± 13.8 ^†^	20.4 ± 13.2
Low resistance	16.9 ± 13.0	17.8 ± 13.2	17.1 ± 12.6

The data are presented as mean ± standard deviation. StO_2_ refers to the oxygen saturation rate, while HbO_2_/HHb represents the relative concentration of oxyhemoglobin/deoxyhemoglobin. The symbol † is used to indicate a statistical difference between low- and high-resistance exercises (*p* < 0.05).

## Data Availability

The data presented in this study are available on request from the corresponding author.

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
