# Peer review of "Myoelectric, Myo-Oxygenation, and Myotonometry Changes during Robot-Assisted Bilateral Arm Exercises with Varying Resistances"

_sensors, 2024, doi:10.3390/s24041061_

Round 1

Reviewer 1 Report

Comments and Suggestions for Authors

The authors conducted robot-assisted bimanual wrist exercise tests. Then, they studied the effects of low/high-resistance exercises using EMG signal, near-infrared spectroscopy, heart rate, and the 25 Borg Rating of Perceived Exertion scale. Then, they compared the two types of exercise. This study is acceptable after the following issues are addressed:

- Adding a paragraph summarizing each section of the paper at the end of the introduction is recommended.

- A comma should be added at the end of the equation in line 205.

- The definition of MF was provided twice (in line 57 and 349). The second definition should be deleted.

- The major contribution of this study is the effects of the exercises observed from the experiments. Thus, it is better to highlight the effects in the discussion with bullet symbols.

- The conclusion simply summarized the results. The authors need to provide limitations of the current study and potential future works.

- The term “we” was frequently used in the manuscript, which is inappropriate in journal articles. This should be removed from the sentences.

- The main contribution of this study is to observe the “fatigue level” of the robot-assisted bimanual upper-limb exercises. Although the introduction mentioned this, the contribution is not noticeable enough. The authors need to highlight the contribution more in the introduction.

- In Figure 2, why were the first 5 and 10 minutes of EMG data used for regression?

Author Response

We sincerely appreciate your insightful comments and suggestions on our manuscript. Your constructive suggestions have been integrated into the document, with the changes visually highlighted in blue within the revised manuscript. In the attached file, we outline the principal modifications made to the manuscript and provide responses to each of your comments.

Reviewer 2 Report

Comments and Suggestions for Authors

This study investigated the physiological responses to fatigue during robot-assisted wrist exercises under two resistance levels: high and low. The researchers employed various techniques to assess muscle fatigue, oxygenation, and recovery, providing valuable insights into the contrasting effects of exercise intensity.

Muscle Fatigue: High-resistance exercise led to a steeper decline in EMG median frequency, indicating greater muscle fatigue compared to low-resistance exercise. This suggests that high-resistance exercise recruits more fast-twitch muscle fibers, which fatigue more readily.

Oxygen Demand: High-resistance exercise also necessitated increased oxygen delivery to the working muscles, as evidenced by higher relative concentrations of oxyhemoglobin and deoxyhemoglobin. This suggests a higher metabolic demand during high-intensity exercise.

Muscle Stiffness: Interestingly, low-resistance exercise led to reductions in muscle stiffness and increased elasticity shortly after exercise and during post-tests. This suggests that light exercise might be beneficial for promoting muscle relaxation and flexibility.

The results highlight the importance of considering exercise intensity when designing robot-assisted rehabilitation programs. While high-resistance exercise can induce greater muscle fatigue and metabolic demand, it may not be the optimal choice for everyone. Low-resistance exercise offers advantages by reducing muscle stiffness, enhancing elasticity, and minimizing both physical and mental exertion.

Investigate the long-term effects of different exercise intensities on muscle strength, function, and recovery in various populations. Explore personalized exercise protocols that optimize training benefits while minimizing fatigue and discomfort based on individual needs and fitness levels. Combine robot-assisted exercises with other interventions like neuromuscular electrical stimulation to further enhance rehabilitation outcomes.

Overall, this study provides valuable evidence for tailoring robot-assisted wrist exercises based on individual goals and preferences. By understanding the physiological responses to different exercise intensities, we can design more effective and personalized rehabilitation programs for improved wrist function and overall well-being.

Author Response

(The authors gave the same response as above.)

Reviewer 3 Report

Comments and Suggestions for Authors

Raws 108-109: For the low-resistance exercise, the active power, inward power, and outward 108 power were set at 40, 50, and 50, respectively, whereas for the high-resistance exercise, 109 these values were adjusted to 80, 50, and 50, respectively:

Can you please better describe what do these values mean? In this way, even who is not familiar with Bi-manu-track can understand in an easier way why you selected these parameters.

Raws 137-138: The EMG from the right extensor carpi 137 radialis and right flexor carpi radialis were then captured

Can you please explain how you choose the electrodes positioning to record muscles’ activity during exercise (surface anatomy, SENIAM guidelines? Etc)

Rwas 150-151: Tissue oxygenation levels in the left extensor carpi radialis and left flexor carpi radialis were assessed using two portable dual-wavelength near-infrared spectroscopy de-151 vices operating at 760 and 850 nm

Can you please explain how you choose the NIRS probes positioning to record muscles’ activity during exercise (surface anatomy, SENIAM guidelines? Etc)

Raws 167-168: to the central region of the left extensor carpi radialis and the left flexor carpi radialis individually, using the MyotonePRO device

Can you please explain how you selected the right position of the Myoton probe on the central part of the muscle belly?

Author Response

(The authors gave the same response as above.)

Reviewer 4 Report

Comments and Suggestions for Authors

The paper provides an investigation into the effects of two types of robot-assisted bimanual wrist exercises on healthy adults. The contrast between long-duration, low-resistance workouts and short-duration, high-resistance exercises is explored, with a focus on assessing fatigue levels through various measures. The inclusion of surface electromyograms, near-infrared spectroscopy, heart rate, and the Borg Rating of Perceived Exertion scale adds depth to the evaluation. The high-resistance exercises demonstrated a more significant decline in electromyogram median frequency and increased hemoglobin recruitment, suggesting heightened muscle fatigue and metabolic demand. Additionally, high-resistance exercises led to increased sympathetic activation and a greater sense of exertion. On the other hand, low-resistance exercises proved advantageous in reducing post-exercise muscle stiffness and enhancing muscle elasticity. The results effectively differentiate between the impacts of low-resistance and high-resistance exercises, providing valuable insights into their physiological effects. The increased muscle fatigue and higher metabolic demand associated with high-resistance exercises are highlighted, along with the heightened sympathetic activation and greater perceived exertion. Conversely, the benefits of low-resistance exercises in reducing post-exercise muscle stiffness and improving muscle elasticity are underscored. The suggestion to opt for a low-resistance setting in robot-assisted wrist movements, given its advantages in alleviating mental and physiological loads, is a practical takeaway.

The study seems interesting but an article proposing an experimental robot-based rehabilitation protocol without any description of the robot in question seems very strange.

The introduction must be rewritten to show the relevance of the problem studied and above all to highlight the contribution of the proposed work.

A recent work presented an intelligent protocol for wrist rehabilitation, could you discuss your proposed protcol compared to the following one 

Bouteraa, Yassine, Ismail Ben Abdallah, and Khalil Boukthir. "A New Wrist–Forearm Rehabilitation Protocol Integrating Human Biomechanics and SVM-Based Machine Learning for Muscle Fatigue Estimation." Bioengineering 10.2 (2023): 219.

The authors report on sixteen healthy right-handed participants. I wonder if all patients have a left hand problem?

The robot cannot serve the right hand? Is it just for left hand rehabilitation?

Could you please give more explanation about the active power, inward power, and outward and about their setting?

Why there is no photos about the different equipment’s used in the experiments?

To improve engagement and motivation, a computer game involving the task of picking up and placing apples was used. I wonder if this game is developed by the authors like the game presented in the following paper 

Bouteraa, Y., Abdallah, I. B., & Elmogy, A. M. (2019). Training of hand rehabilitation using low cost exoskeleton and vision-based game interface. Journal of Intelligent & Robotic Systems, 96, 31-47.

We need also more explanation in this context?

The EMG signals underwent two filtering processes. Where these filters were implemented and how?

Overall, many explanations are necessary, whether at the level of material and methods or at the level of results and discussions.

Comments on the Quality of English Language

English written needs to be polished further

Author Response

(The authors gave the same response as above.)

Reviewer 5 Report

Comments and Suggestions for Authors

- The authors should provide a more detailed explanation and justification for the specific choices of myotonometric parameters (oscillatory frequency, decrement, stiffness), and discuss the clinical significance of changes in these parameters in the context of pronation and supination;

- Please include a more detailed explanation of the clinical significance of the observed changes in EMG median frequency is crucial, including how these changes might influence performance and recovery;

While the study mentions an increase in Borg RPE and mean heart rate, it would be beneficial to discuss the specific values and ranges observed at different stages of both low-resistance and high-resistance exercises. This information could provide a more nuanced understanding of the perceived exertion and physiological responses throughout the exercise sessions. To enhance the discussion, consider integrating the Borg RPE scores with physiological responses such as EMG, near-infrared spectroscopy, and myotonometry. This could provide a comprehensive view of how perceived exertion aligns with objective physiological measurements, offering valuable insights into the relationship between subjective and objective indicators of fatigue;

While the study concludes with general statements about the advantages of low-resistance exercises, consider expanding on the practical implications of the findings. How can this information be applied in rehabilitation or training programs?

Author Response

(The authors gave the same response as above.)

Round 2

Reviewer 4 Report

Comments and Suggestions for Authors

 I would like to express my sincere appreciation for your dedication and efforts in revising your paper.